# The Kinetics of FMS-Related Tyrosine Kinase 3 Ligand (Flt-3L) during Chemoradiotherapy Suggests a Potential Gain from the Earlier Initiation of Immunotherapy

**DOI:** 10.3390/cancers14163844

**Published:** 2022-08-09

**Authors:** Łukasz Kuncman, Magdalena Orzechowska, Konrad Stawiski, Michał Masłowski, Magdalena Ciążyńska, Leszek Gottwald, Tomasz Milecki, Jacek Fijuth

**Affiliations:** 1Department of Radiotherapy, Medical University of Lodz, 90-419 Lodz, Poland; 2Department of Molecular Carcinogenesis, Medical University of Lodz, 90-419 Lodz, Poland; 3Department of Biostatistics and Translational Medicine, Medical University of Lodz, 90-419 Lodz, Poland; 4Department of External Beam Radiotherapy, Regional Cancer Center, Copernicus Memorial Hospital of Lodz, 93-513 Lodz, Poland; 5Department of Dermatology, Paediatric Dermatology and Oncology Clinic, Medical University of Lodz, 90-419 Lodz, Poland; 6Department of Urology, Poznan University of Medical Sciences, 61-701 Poznan, Poland

**Keywords:** chemoradiotherapy, immunotherapy, radiotherapy, FMS-related tyrosine kinase 3 ligand (Flt-3L), dendric cells, active bone marrow, lymphopenia

## Abstract

**Simple Summary:**

Combining chemoradiotherapy with immunotherapy is one of the main milestones in cancer treatment. The optimal sequence to ensure the effectiveness of the treatment is unknown. We investigated the kinetics of FMS-related tyrosine kinase 3 ligand (Flt-3L), a multi-potential haemopoietic protein, and observed an increase in its concentration following bone marrow-damaging irradiation. At the same time, Flt-3L is an important regulating factor of dendritic cells, which are essential for the immune effect of lymphocytes. We observed an increase in Flt-3L levels in the first two weeks of treatment with no further increase during chemoradiotherapy. Our model explains the early variability of Flt-3L, which was related to the irradiation dose in active bone marrow and the subsequent lymphopenia. Our results argue for the earlier initiation of immunotherapy when the concentration of Flt-3L is high and no lymphopenia has yet occurred.

**Abstract:**

The optimal sequence of chemoradiotherapy with immunotherapy is still not established. The patient’s immune status may play a role in determining this order. We aim to determine the kinetics of a multi-potential haemopoietic factor FMS-related tyrosine kinase 3 ligand (Flt-3L) during chemoradiotherapy. Our pilot, a single arm prospective study, enrolled patients with rectal cancer who qualified for neoadjuvant chemoradiotherapy. Blood samples for Flt-3L were collected before and every second week of chemoradiotherapy for a complete blood count every week. The kinetics of Flt-3L were assessed using Friedman’s ANOVA. A multiple factor analysis (MFA) was performed to find relevant factors affecting levels of serum Flt-3L during chemoradiotherapy. FactoMineR and factoextra R packages were used for analysis. In the 33 patients enrolled, the level of Flt-3L increased from the second week and remained elevated until the end of treatment (*p* < 0.01). All patients experienced Grade ≥2 lymphopenia with a nadir detected mostly in the 5/6th week. MFA revealed the spatial partitioning of patients among the first and second dimensions (explained by 38.49% and 23.14% variance). The distribution along these dimensions represents the magnitude of early changes of Flt-3L. Patients with the lowest values of Flt-3L change showed the highest lymphocyte nadirs and lowest dose/volume parameters of active bone marrow. Our hypothesis-generating study supports the concept of early initiation of immuno-therapy when the concentration of Flt-3L is high and no lymphopenia has yet occurred.

## 1. Introduction

Currently, most immunotherapy-based oncology treatments are based on inhibition of the PD1/PD-L1 and CTLA4 axes and result in the stimulation of T lymphocytes [1]. Although most immunotherapy is focused on lymphocytes, dendritic cells (DC) within the tumor microenvironment are required for their activation. They play the role of decision-making nodes [2]. One of their main roles is to initiate the immune response by cross-presenting antigens to T lymphocytes [3]. Immune tumor response is due to the endogenous secretion of interferon (IFN) β which activates tumor-infiltrating dendric cells following cellular DNA recognition (by dendritic cells) via the cGAS-STING pathway [4]. Similarly, radiotherapy, by damaging cells and leading to the release of danger/damage associated molecular patterns (DAMPs), including cytosolic DNA, stimulates the activity of dendritic cells through the cGAP-STING pathway [5]. The similarity of the above signaling pathways makes radiation therapy a promising partner for dendritic cell immunotherapy and may overcome resistance to PD-L1 therapy in poorly immunogenic tumors, which has been shown in preclinical studies [6].

The FMS-related tyrosine kinase 3 ligand (Flt-3L) is a transmembrane protein that can be secreted as a soluble protein [7,8]. It is an important regulator of hematopoiesis and has a pivotal role in dendric cell activation [7,8]. Flt-3L acts through the FMS-related tyrosine kinase 3(Flt-3) present on the multipotent progenitors and common lymphoid progenitors as their growth factor but not on mature cells, apart from DC [2]. The function of the Flt-3L/Flt-3 axis is important in both the development and function of mature plasmacytoid dendric cells (DC) (pDC) and conventional DC (cDC), which has been proven in mice models lacking the corresponding genes [9].

The increase in the concentration of Flt-3L as a main regulator of hemopoiesis is observed after bone marrow damaging factors, such as chemotherapy and radiotherapy [10,11]. The Flt-3L level was dependent on the radiation dose and the irradiation area in the mouse model [12]. Preclinical studies also suggest that Flt-3L may be involved in the radio-resistance of cancer cells [7]. Although numerous preclinical data show an association of Flt-3L with radiation therapy, there is no significant clinical evidence regarding the kinetics of Flt-3L during fractionated chemoradiotherapy in humans.

Faced with the growing role of the combination of chemoradiotherapy and immunotherapy, we designed a pilot study to assess the kinetics of Flt-3L and clinical factors triggering the concentration of Flt-3L in a homogeneous cohort of patients undergoing modern planned 3-D chemoradiotherapy with an assessment of dose volume parameters in the active bone marrow.

## 2. Materials and Methods

### 2.1. Study Design and Participants

We conducted a single-center, single-arm prospective cohort study. Patients aged over 18 years, with histologically confirmed adenocarcinoma of the rectum, UICC (7th edition) clinical stage II and III, and who were qualified to neoadjuvant chemoradiotherapy were eligible for enrolment to this study. Patients had an Eastern Cooperative Oncology Group (ECOG) status score of 0–2 and no serious hematologic, renal, liver, or cardiac disorders. Key exclusion criteria were previous systemic oncological treatment, radiotherapy of the pelvis region, systemic diseases of the intestines, and inflammation of the pelvis. All patients provided written informed consent. The study protocol was approved by the Medical University of Lodz Ethics Committee (RNN/53/16/KE). The study was carried out in accordance with the provisions of the Declaration of Helsinki and Good Clinical Practice guidelines.

### 2.2. Procedures

Patients were qualified to neoadjuvant chemoradiotherapy based on the decision of the rectal multidisciplinary team according to the current National Comprehensive Cancer Network (NCCN) rectal cancer guidelines. All patients underwent radiotherapy planning computer tomography (CT) and 1.5T magnetic resonance imaging (MRI) with T1-weighted sequence encompassing all pelvis region bones. Three-Dimensional Conformal Radiation Therapy (3D-CRT) or dynamic irradiation techniques (Intensity Modulated Radiation Therapy (IMRT)/Volumetric Modulated Arc Therapy (VMAT)) were used to treat patients to 50.4–54 Gy in 1.8 Gy fractions. The delineation of radiotherapy clinical target volumes (CTV) were carried out according to Radiation Therapy Oncology Group consensus [13]. A two-day 5-fluorouracil (400 mg/m^2^/d) with leucovorin (20 mg/m^2^/d) regimen was prescribed concurrently with radiotherapy every 14 days. On the basis of the T1-weighted sequence of MRI active bone marrow (BMact) and total bone marrow (BMtot) in pelvis region was delineated as described in our previous study [14]. The radiotherapy treatment plan was prepared in the Eclipse (Varian USA) system, in the version that was current at the time of treatment preparation.

Blood samples for Flt-3L testing were collected on the first day, before the start of chemoradiotherapy(Flt-3L 0), after two (Flt-3L 2) and four (Flt-3L 4) weeks, each time before the administration of chemotherapy, and in the 5/6th (Flt-3L 5/6) week at the end of treatment. Blood samples were centrifuged immediately after collection. Plasma was stored at a temperature below −20 °C. The determination of the plasma concentration of Flt-3L was performed after the completion of blood sample collection, using Elisa R&D Systems (Minneapolis, MN, USA) kit no. DFK00 and in accordance with the manufacturer’s instruction by university laboratory with experience in the method.

According to department policy, blood was collected for a complete blood count every week for biochemical examination prior to the administration of chemotherapy.

### 2.3. Endpoints and Statistical Analysis

The required number of patients has been estimated prospectively as described in our previous study [14]. The Flt-3L kinetics analysis was divided into two stages. 

The endpoint was the assessment of whether and how the concentration of Flt-3L in the blood varies with time during fractionated chemoradiotherapy in a homogeneous group of patients in terms of diagnosis and treatment. The assessment of the variability of the Flt-3L concentration over time was performed using Friedman’s ANOVA followed by post hoc analysis by Wilcoxon’s signed-rank tests with Bonferroni’s correction due to the expected lack of a normal data distribution and the size of the patient group.

The second goal of the study, in the case of confirmation of the first endpoint, was to assess the clinical usefulness of the Flt-3L level. Multiple Factor Analysis by Escofier and Pages (MFA) based on Principal Component Analysis (PCA) was performed to find the relevant clinical factors affecting the levels of serum Flt-3L during chemoradiotherapy [15,16]. The analysis was performed using FactoMineR and factoextra R packages [17].

## 3. Results

### 3.1. Clinical Data

Thirty-five patients were enrolled in the study and two did not have all the scheduled Flt-3L tests collected due to withdrawal of consent. All participants (16 female, 17 male) aged 44–76 years were in good condition (ECOG 0–16, ECOG 1–7) and had completed scheduled courses of chemoradiotherapy in 37 to 44 days. One patient did not receive half of the second course of chemotherapy because of an infection. Steroids were not administered during the study. All patients experienced Grade ≥2 lymphopenia according to Common Terminology Criteria for Adverse Events (CTCAE) v4.0 (grade 3–78.8% of patients, grade 2–21.2%). A lymphocyte nadir occurred in the 6th week of chemoradiotherapy in 42.4% of the patients, in the 5th week in 39.4% of the patients, in the 4th week in 15.2% of patients, in the 3rd week of in 3% of patients. The lymphocyte nadir did not occur in any of the patients in the 1st or 2nd week of chemoradiotherapy. No other grade ≥2 CTCAE v4.0 hematologic toxicity occurred.

### 3.2. Flt-3L Kinetics

The median concentration of Flt-3L 0 was 79.55 pg/mL [55.66–132.47 pg/mL], the median Flt-3L 2 was 127.93 pg/mL [62.94–363.41 pg/mL], the median Flt-3L 4 was 124.65 pg/mL [11.62–194.80 pg/mL], and the median Flt-3L 5/6 was 124.29 pg/mL [56.15–187.77 pg/mL]. The concentration of Flt-3L in blood increased from the beginning of radiotherapy and remained above the nadir until the end of radiotherapy in all but one patient. The concentration of Flt-3L 0 was significantly different from Flt-3L 2, Flt-3L 4, and Flt-3L 6 (*p* < 0.01). There were no statistically significant differences between the concentrations of Flt-3L 2, Flt-3L 4, and Flt-3L 6. The kinetics of the Flt-3L concentration are shown in Figure 1, with the Wilcoxon’s rank tests post hoc analysis of Flt-3L 0, Flt-3L 2, Flt-3L 4, and Flt-3L 5/6 values shown in Table 1.

### 3.3. Multiple Factor Analysis (MFA)

Primarily, we applied the MFA to reduce the clinical variables to only those that were associated with the hematological toxicity of chemoradiotherapy as possibly important for the early growth of Flt-3L. Afterwards, the multidimensional model included the platelet toxicity, lymphocyte toxicity, gender, and dosimetry data of active bone marrow as well as the concentration of Flt-3L. We observed the spatial partitioning of patients among the first (explained by 38.49% variance) and second dimensions (explained by 23.14% variance). According to the correlation circle (Figure 2), dimension 1 was positively correlated with the lymphocyte nadir and simultaneously negatively correlated with the dose/volume data of active bone marrow in the low to high doses range (V5BMact to V45Bmact-volumes of active bone marrow receiving doses from 5 Gy to 45 Gy, respectively). However, none of these parameters played a dominant role. In turn, dimension 2 was mostly correlated with the first and second measurement of Flt-3L and partially with platelet nadir. An individual factor map is shown in Figure 3. Further investigations of the model revealed that the distribution of samples along dimensions 1 and 2 represents the magnitude of the Flt-3L delta (herein as the difference between measurements of Flt-3L 0 and Flt-3L 2). Figure 4 combined with Figure 2 show that the patients with the lowest values of Flt-3L delta simultaneously showed the highest lymphocyte and platelet nadirs, whereas the patients with the highest values of Flt-3L delta were related to the highest dose/volume parameters of active bone marrow. We also noticed disparities in the aforementioned parameters according to sex (Figure 5). Females had lower lymphocyte and platelet nadirs with higher dose/volume parameters of active bone marrow and, most importantly, the largest increase in Flt-3L.

## 4. Discussion

The combination of chemoradiotherapy and consolidating immunotherapy is a well-established treatment option in patients with lung cancer and one of the recent breakthroughs in radical treatment [18]. In other cancers, such as rectal cancer, we are awaiting the results of clinical trials [19]. Understanding the immune mechanisms during chemoimmunotherapy may be crucial in selecting the optimal treatment sequence and increasing its effectiveness. Rectal cancer is of particular interest in this context. Untreated it is non-immunogenic, and the benefit of immunotherapy is questionable [19]. The use of chemoradiotherapy/radiotherapy drastically changes the tumor microenvironment and may increase the tumor’s immunogenicity, largely driven by the activation of DCs [19,20]. Irradiated tumor cells undergo immunogenic cell death, which leads to the release of DAMPs and an increase in the concentration of cytokines, such as IFN-γ, IL-1, TGF-β, and IL-6, followed by the stimulation of DCs [19,21]. The Flt-3/Flt-3L signal path plays an important role in the function of DCs and the primary data indicate the possibility of modulating the effect of DCs with drugs based on Flt-3 [2,22]. The Flt-3/Flt-3L signal path has been gaining importance recently in the treatment of both solid tumors and leukemias [2]. In acute myeloid leukemia it has already changed the landscape of treatment in first-line and salvage settings, becoming the standard of care in Flt-3 mutated leukemias [23]. In solid tumors, clinical trials are ongoing, including those involving the combination of Flt-3L with radiotherapy [2,24].

The dual function of Flt-3L as a multi-potential haemopoietic factor and a boost factor for DCs is particularly important in combination with chemoradiotherapy/radiotherapy due to the similar signaling pathways described above [2]. When designing the study, we wanted to evaluate the kinetics of Flt-3L in a homogeneous group of patients in terms of the dose of radiation and applied systemic treatment. Almost half of the bone marrow is located in the pelvis, hence the assessment of Flt-3L during chemoradiotherapy in this region seems to be the most rational [25]. In preclinical studies, the blood concentration of Flt-3L was the most important predictor of irradiation of the whole or a part of the body on the 1st, 3rd, and 7th day after irradiation [26]. Additionally, chemotherapy used in the course of chemoradiotherapy for rectal cancer has a low myelotoxic potential, which allows for a better assessment of the effects of radiotherapy [27].

We observed an increase in Flt-3L concentration from the second week of chemoradiotherapy, which remained elevated until the end of chemoradiotherapy. To the best of our knowledge, a similar study on a homogeneous group of patients, assessing the kinetics of Flt-3L during 3D planned chemoradiation therapy, has not been carried out. The only clinical study by Huchet et al. showed a dependence of Flt-3L on the dose in the bone marrow determined on the basis of nomograms and Flt-3L inversely correlated with the level of white blood cells and platelets [28]. Conclusions about the kinetics of Flt-3L are difficult to draw because the study used different radiotherapy regimens (probably 2D planned) in patients with different tumor locations, and some of them used steroids [28]. More data comes from preclinical studies. After the single-fraction (1–8 Gy) irradiation of parts or the whole body of mice, the Flt-3L concentration increased dose-dependently 72h after irradiation and decreased 3 weeks after RT [12]. The irradiation of the trunk, where most of the bone marrow is located, was associated with an increase in the concentration of Flt-3L, in contrast to the irradiation of the limbs and head [12]. Similarly, in another study, after the irradiation of 25, 50, 75, or 100% of the bone marrow in mice with doses of 4, 7.5, or 11 Gy, the concentration of Flt-3L correlated with the volume of irradiated marrow and the severity of pancytopenia [29]. Our results are in line with above preclinical studies but more clinically useful. When assessing the kinetics of Flt-3L, the first change (Flt-3L delta) may be the most relevant as it allows for the possible modification of further treatment. In our study, separation in MFA matrix, according to Flt-3L delta, was associated with active bone marrow dose–volume parameters and lymphocytes and platelets nadirs.

In our study, Flt-3L, as a haemopoietic protein, showed the greatest increase in patients with higher doses in active bone marrow, as well as those who developed lymphopenia. The haemopoietic potential driven by Flt-3L increases to the 2nd week of treatment and remains at an elevated level to the end of chemoradiotherapy. In addition, lymphopenia occurs at the end of treatment, and in our study 81.8% of patients developed a lymphocyte nadir in the 5th or 6th week of chemoradiation therapy. Lymphopenia is a critically important factor influencing the effectiveness of immunotherapy and the survival of patients being treated with chemoradiotherapy combined with immunotherapy [30,31]. This seems to be clinically relevant in the context of discussions about the optimal timing of immunotherapy. The study by Kim et al. brings similar data about the dynamics of immune cells during chemoradiotherapy [32]. The lymphopenia was often observed with the robust proliferation of CD8+ lymphocytes during chemoradiotherapy [32]. Additionally, the change in the phenotype of lymphocytes indicated a worse response to immunotherapy after the end of chemoradiotherapy [32]. The optimal sequence of chemoradiotherapy and immunotherapy has not been established, and the simultaneous use of chemoradiotherapy and immunotherapy seems to be possible in phase II studies [33]. Our study supports early the administration of immunotherapy (at the beginning of chemoradiotherapy) when the blood levels of Flt-3L are high and dendritic cells may be affected and before lymphopenia occurs.

We also observed separation between sexes in the MFA analysis. This could be due to the higher dose in the bone marrow associated with another pelvic anatomy, as previously described [14,34]. The difference may also result from the different tolerance of systemic treatment between women and men [35].

Our trial has some limitations; it was conducted using a relatively small sample size, but the number of patients was accessed prospectively and is comparable to another similar study [32]. Multiple factorial analysis is a powerful extension of principal component analysis but is dedicated for exploratory purposes and hypothesis generation, which is in line with our study design.

Despite the above limitations, our study has strong points. To our knowledge, it is the first to assess Flt-3L kinetics during chemoradiotherapy. It was a prospective study carried out on a homogeneous group of patients in terms of diagnosis, tumor stage, general condition, and treatment method. Additionally, we used dose/volume parameters in the active bone marrow, which is a more accurate method compared to the whole bone marrow.

## 5. Conclusions

The concentration of Flt-3L in the peripheral blood increases in the first two weeks of chemoradiotherapy and remains at a similar level until the end of chemoradiotherapy. The amount of the initial increase in Flt-3L concentration was associated with higher dose/volume parameters of active bone marrow and inversely with lymphopenia. Our hypothesis-generating study supports the concept of early initiation of immunotherapy when the concentration of Flt-3L is high and no lymphopenia has yet occurred.

## Figures and Tables

**Figure 1 cancers-14-03844-f001:**
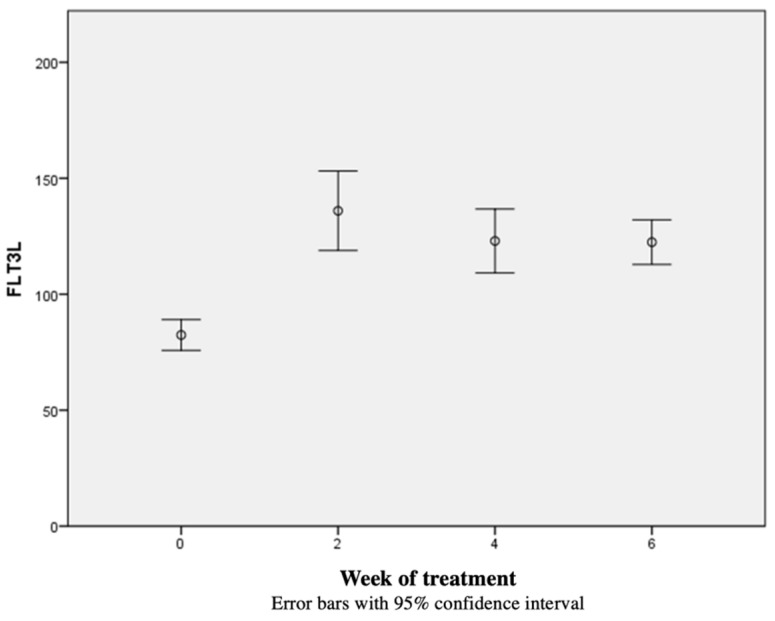
Kinetics of Flt-3L in blood shown as a result of Friedman’s ANOVA test. The *y* axis shows the blood concentration of Flt-3L (pg/mL), the *x* axis shows the measurement time (weeks).

**Figure 2 cancers-14-03844-f002:**
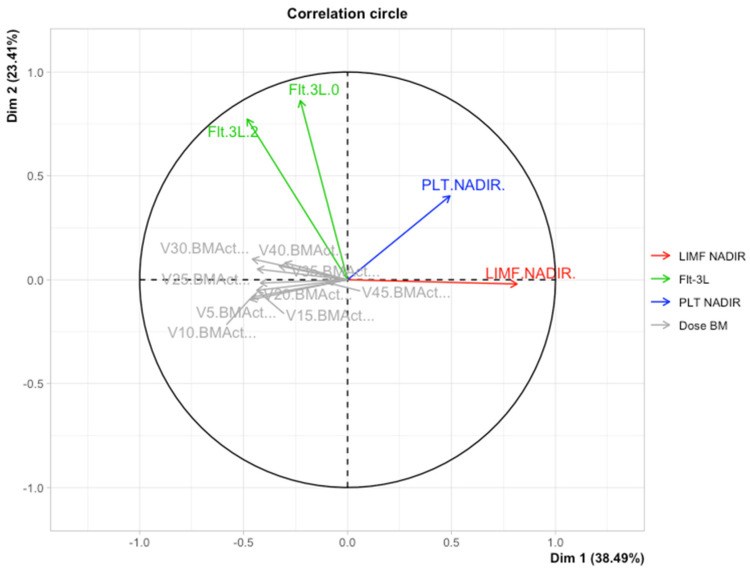
Multiple factorial analysis correlation circle of dose volume parameters, platelet and lymphocyte nadirs, and Flt-3L concentration. Dimension 1 (explained by 38.49% variance) shown on the *x*-axis, dimension 2 (explained by 23.14% variance) shown on the *y*-axis.

**Figure 3 cancers-14-03844-f003:**
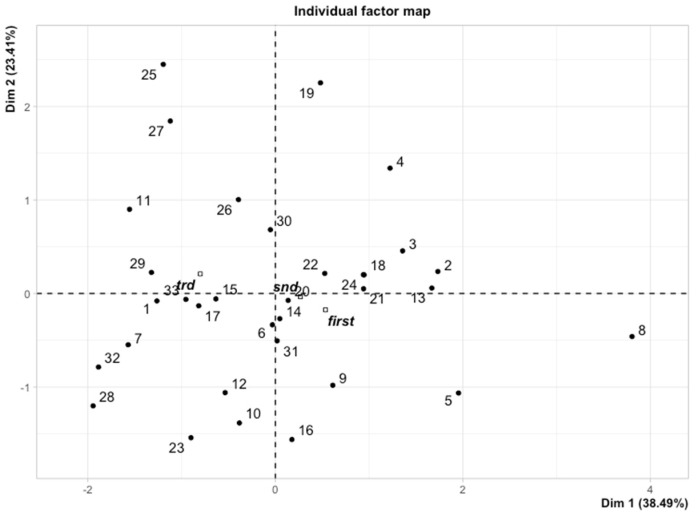
Individual factor map showing the position of each patient in the dimension 1 and 2 matrix.

**Figure 4 cancers-14-03844-f004:**
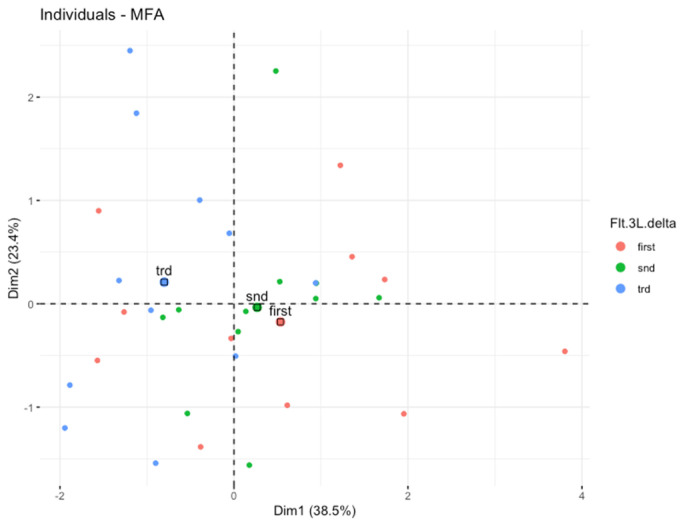
Individual factor map showing data distribution. According to early change of Flt-3L (Flt-3L delta).

**Figure 5 cancers-14-03844-f005:**
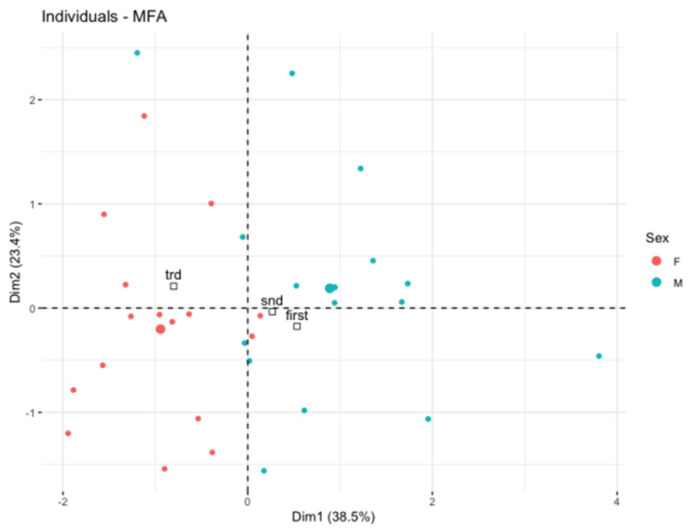
Individual factor map showing gender-based data distribution. F—Female individuals colored in red, M—Male individuals colored in blue.

**Table 1 cancers-14-03844-t001:** Comparison of Flt-3L blood concentration over time. Results of Wilcoxon rank tests with post hoc analysis of Flt-3L 0, Flt-3L 2, Flt-3L 4, and Flt-3L 5/6. Statistically significant data has been shaded.

Data	Flt-3L 2 and Flt-3L 0	Flt-3L 4 and Flt-3L 0	Flt-3L 5/6 and Flt-3L 0	Flt-3L 4 and Flt-3L 2	Flt-3L 5/6 and Flt-3L 2	Flt-3L 5/6 and Flt-3L 4
Z	−4.91	−4.08	−4.58	−0.49	−1.37	−0.94
Bilateral asymptomatic significance	<0.01	<0.01	<0.01	NS	NS	NS

## Data Availability

The data presented in this study are openly available at https://1drv.ms/x/s!AqwHYmZlPESTgoYv95vIMOmTrVr3mg?e=MO2YnR (accessed on 8 August 2022).

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
