# Peer review of "The Kinetics of FMS-Related Tyrosine Kinase 3 Ligand (Flt-3L) during Chemoradiotherapy Suggests a Potential Gain from the Earlier Initiation of Immunotherapy"

_cancers, 2022, doi:10.3390/cancers14163844_

Round 1

Reviewer 1 Report

The authors address a pressing question of when to integrate immune checkpoint inhibitors with RT. The presented dynamic tumor biology data of FLT-3 levels during standard concurrent chemoradiotherapy gives insight to the early use of immune checkpoint inhibitors concurrent with RT. As with all of the evolving clinical utility and use of immune checkpoint inhibitors in earlier stage cancers, every piece of data is clinically helpful. This study adds to that knowledge.

Author Response

Dear Reviewer
Thank you for your time and positive evaluation of our manuscript.

Reviewer 2 Report

I appreciate the initiative of the research team and the concern to contribute to the elucidation of the optimal moment in which the combination of chemoradiotherapy with immunotherapy is beneficial for patients with neoplasia and especially for those with rectal cancer after radiotherapy who qualify for neoadjuvant chemoradiotherapy. In addition, the choice to evaluate Flt-3L kinetics during chemoradiotherapy is innovative and adds value to this prospective study.

Your study is particularly complex and I understand the difficulties that arise during its development.

It would be useful to explain why the variability of Flt-3L was assessed by Friedman's ANOVA followed by post-hoc analysis by Wilcoxon signed-rank tests with Bonferroni's correction.

I also consider it useful for the figures to be more clearly explained and the parameters in the graph to have more explicit legends.

I understand that your study supports early administration of immunotherapy when blood levels of Flt-3L are high and dendritic cells may be affected and before lymphopenia occurs but if there were patients in whom Flt-3L is low and associates lymphopenia or there are patients in whom it occurs however, lymphopenia simultaneously with the increased values ​​of Flt-3L.

Author Response

Response to Reviewer 2 Comments

Dear Reviewer,

We appreciate the time and effort that you have dedicated to providing your valuable feedback on our manuscript. We have modified the paper in response to your comments. We have addressed each of your comments as outlined below, corrections in manuscript text have been made in “tracking changes mode” in MS Office Word. Additionally, the article has been revised by the native speaker.       

Response 1: It would be useful to explain why the variability of Flt-3L was assessed by Friedman's ANOVA followed by post-hoc analysis by Wilcoxon signed-rank tests with Bonferroni's correction.”

Thanks for your comment. We have made a refinement in the content of the article (line 235-236). Friedman's ANOVA followed by post-hoc analysis by Wilcoxon signed-rank tests is a non-parametric test used to evaluate the variability of non-normally distributed data. The reliability of Friedman's ANOVA for small groups of patients is much more reliable than its parametric counterpart ANOVA. Additionally, we used the Bonferroni correction to further reduce the uncertainty of the results. The tests were selected prospectively and, in the opinion of two co-authors with biostatistician education, were the most reliable for the evaluation of our data.

Response 2: I also consider it useful for the figures to be more clearly explained and the parameters in the graph to have more explicit legends.”

Thank you for your suggestion, we have included a more detailed explanation of figures. Modification have been made in manuscript text.

Response 3 “I understand that your study supports early administration of immunotherapy when blood levels of Flt-3L are high and dendritic cells may be affected and before lymphopenia occurs but if there were patients in whom Flt-3L is low and associates lymphopenia or there are patients in whom it occurs however, lymphopenia simultaneously with the increased values of Flt-3L.”

Thank you for your valuable comment/question. None of the patients had a lymphocyte nadir at baseline (before treatment), during the first or the second week of chemoradaiotehrapy, only 3% in third week. At the same time, blood levels of Flt-3L in almost all (except one patient) increased from the second week of chemoradiotherapy. Described by you situation didn’t occurred in our cohort. We are aware that our study is a pilot study and we are unable to draw conclusions about the reasons of late Flt-3L concentration increased in one patient.

We have described the time of nadir occurrence in lines 253-254. We have modified the text based on your comment(lines 261-262) to make it clear for readers.
